# Study on Verification Approach and Multicontact Points Issue When Modeling *Cyperus esculentus* Seeds Based on DEM

**Tianyue Xu** [1] , **Ruxin Zhang** [1], **Xinming Jiang** [1], **Weizhi Feng** [1], **Yang Wang** [2] **and Jingli Wang** [1,*]

1    College of Engineering and Technology, Jilin Agricultural University, Changchun 130118, China
2    College of Biological and Agricultural Engineering, Jilin University, Changchun 130021, China
*    Correspondence: wjlwy2004@sina.com; Tel.: +86-1350-0819-883

**Abstract:** In this paper, the Multisphere (MS) models of three varieties of *Cyperus esculentus* seeds are modeled based on DEM. In addition, for comparison, other particle models based on automatic filing in EDEM software are also introduced. Then, the direct shear test, piling test, bulk density test, and rotating hub test are used to verify the feasibility of particle models of *Cyperus esculentus* seeds that we proposed. By comparing the simulated results and experimental results, combined with the CPU computation time, the proposed particle models achieved better simulation accuracy with fewer filing spheres. According to simulation results, some limitation was present when using one single verification test; varieties of verification tests used could improve the verification reliability, and a more appropriate particle model could be selected. Additionally, the issue of multicontact points in the MS model was studied. The Hertz Mindlin (no slip) (HM) model and Hertz Mindlin new restitution (HMNR) model were both considered in simulations for comparison. The rotating hub test and particle–wall impact test were used, and the influences of multiple contact points on the motion behavior of individual particles and particle assemblies were analyzed. Simulation results showed that the multiple contact points affected the motion behavior of individual particles; in contrast, the influence of multiple contact points on the motion behavior of the particle assembly was insignificant. Moreover, the relationships between moisture content of seeds and Young's modulus, Young's modulus, and the number of contact points were also considered. Young's modulus decreased with increasing moisture content. The number of contact points increased with a decreasing Young's modulus.

**Keywords:** *Cyperus esculentus* seeds; the multisphere (MS) method; verification test; multicontact points; Young's modulus

## 1. Introduction

*Cyperus esculentus* is a type of cash crop, and the cultivated area is increasing yearly. However, the mechanization level is not adequate for needs; thus, it is necessary to develop sowing and harvesting machinery for *Cyperus esculentus*. When designing and optimizing relevant mechanical components, contacts between relative components and the *Cyperus esculentus* seeds occur. To analyze these contacts, an increasing number of scholars have used the discrete element method (DEM) [1–8], which is a type of numerical simulation method that was used to simulate the motion behavior of granular materials. DEM can describe the translation and rotation of particles using Newton's Second Law and Euler's dynamic equation, respectively. DEM also provides a solution when analyzing the complex motion behavior of granular materials.

Additionally, due to the irregular shape of the *Cyperus esculentus* seeds, the contact judgment between seeds is more complex than that between spherical particles. It is difficult to achieve a higher accuracy of force analysis between particles [9]. Recently, the modeling methods for irregular particles can be divided into the super quadratic equation method [10–12] and the MS method [10,13,14]. The computational process of the super

quadratic equation method is more complex; in contrast, the MS method is more intuitive and widely used. In agricultural engineering, an increasing number of researchers have used the MS method to model agricultural granular materials. Xu et al. [15] and Yan et al. [16] modeled soybean seeds using the MS method. Wang et al. [17], Chen et al. [18], and Zhou et al. [19] modeled a corn seed based on the MS method. Sun et al. [20] modeled wheat seeds using the MS method. Many studies have shown that the more filing spheres included, the closer the model to the real seed could be achieved. However, the precision particle shape model could not markedly improve the simulation accuracy. Conversely, the more filing spheres existed in particle modeling, the longer the computation time for contact judgment consumed. Generally, an MS model with fewer spherical elements could obtain a better simulation accuracy [21]. Therefore, how to create a more precise and appropriate DEM model for *Cyperus esculentus* seeds should be studied in more detail. Common verification methods currently include the piling test, bulk density test, and "self-sieving" test [15–20,22]. The issue of multicontact points could be generated when particle modeling is based on the MS method in simulation. The multicontact points could affect granular motion behavior. Kruggel-Emden [23] and Höhner [24] investigated the influence of multicontact points on granular motion behavior through individual particles impacting the boundary. Then, the corresponding solution formula was proposed. Yan et al. [25] suggested that the influence of multiple contact points on the movement of individual particles would not affect the particle assembly motion behavior. Zhou et al. [26] proposed that the influence of multiple contact points on the motion of individual particles was strong via a "self-sieving" test and piling test using the HMNR model. However, for the DEM model of *Cyperus esculentus* created in this study, the issue of whether multicontact points affected the motion behavior of individual particles and particle assemblies must be studied in more detail. In addition, Young's modulus had a strong influence on the multicontact points issue [27,28]. The moisture content of seeds could have a significant effect on Young's modulus of seeds. The relationships between the moisture content of seeds and Young's modulus, Young's modulus, and the number of contact points should be considered. Therefore, how Young's modulus affects the contact points when using *Cyperus esculentus* modeling, which we proposed, must be studied in more detail.

In this study, the DEM models of three varieties of *Cyperus esculentus* seeds were created with different numbers of filing spheres. In addition, for comparison, another particle model based on automatic filing in EDEM software was also introduced. To verify the test with the highest reliability, the direct shear test, piling test, bulk density test, and rotating hub test were used. In addition, the HMNR model can reduce the effect of multicontact points in the simulation when the MS model is used [24,29]. Therefore, the HMNR model and HM model were both used for comparison, and the influences of the multicontact points on the movement behavior of the particle assembly and the individual particle were studied through simulation analysis of the rotating hub test and the free drop test. The rotating hub test, which described the movement of the particle assembly, and the free drop test, which described the movement of individual particles, were both considered. Additionally, the influence of Young's modulus on the multicontact points was also investigated.

## 2. Cyperus Esculentus Seed Modeling

In this study, three varieties of *Cyperus esculentus* seeds were considered, as shown in Figure 1a–c. The density of Jinong 1 was 1.34 g/cm$^3$, with a moisture content of 28.8% and a thousand seed weight of 427 g. The density of Jinong 2 was 1.27 g/cm$^3$, with a moisture content of 28.4% and a thousand seed weight of 406 g. The density of Jinong 3 was 1.19 g/cm$^3$, with a moisture content of 35.8% and a thousand seed weight of 809 g. Additionally, the mean value and standard deviation of the tri-axial dimensions of the *Cyperus esculentus* seeds are shown in Table 1.

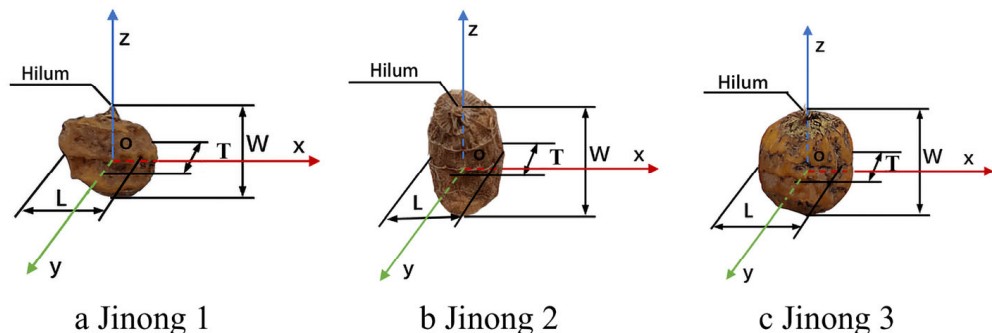

a Jinong 1       b Jinong 2       c Jinong 3

**Figure 1.** (**a**–**c**) Three varieties of *Cyperus esculentus* seeds [22].

**Table 1.** *Cyperus esculentus* average of tri-axial dimensions and standard deviation [22].

| Variety | Size/mm | Mean/mm | Standard Deviation/mm |
|---------|---------|---------|------------------------|
| Jinong 1 | Length (*L*) | 9.63 | 0.54 |
|  | Width (*W*) | 9.11 | 1.19 |
|  | Thickness (*T*) | 7.94 | 1.43 |
| Jinong 2 | Length (*L*) | 8.02 | 0.62 |
|  | Width (*W*) | 13.70 | 1.71 |
|  | Thickness (*T*) | 5.80 | 0.65 |
| Jinong 3 | Length (*L*) | 11.86 | 1.37 |
|  | Width (*W*) | 11.45 | 1.56 |
|  | Thickness (*T*) | 9.49 | 1.56 |

The changes in the tri-axial dimensions and Young's modulus for three varieties of *Cyperus esculentus* seeds under different moisture content were studied. When the indoor temperature was 23 °C, the seeds were immersed in pure water for 0 h, 12 h, 24 h, 36 h, and 48 h, respectively, and then the tri-axial dimensions were measured. The changes in moisture content for different immersed time is shown in Figure 2. It could be seen from the figure that the moisture content of three varieties of the seeds increased with increasing time. Table 2 below shows the changes in the tri-axial dimensions before and after hydroponics. It could be seen that the tri-axial dimensions of seeds increase with the moisture content increasing. After immersing for 48 h, the width of Jinong 2 increased with an increase of 61.84%.

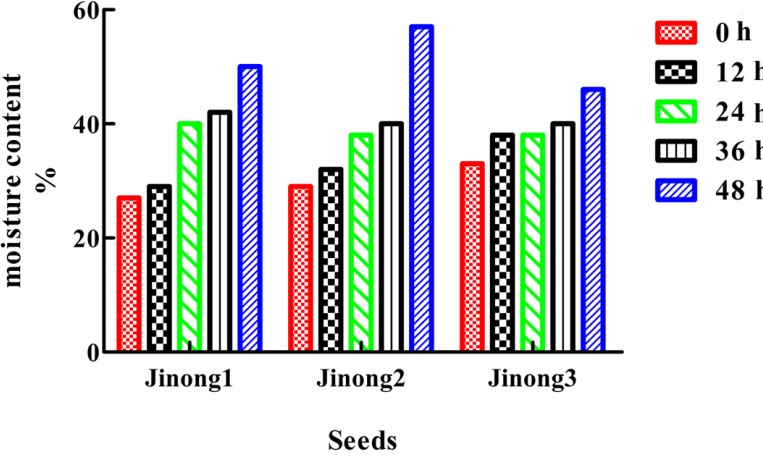

**Figure 2.** The changes in moisture content for different immersed time.

**Table 2.** The changes in tri-axial dimensions for the three varieties of the *Cyperus esculentus* seed before and after hydroponics.

| Time | Type | Size Before Hydroponic Culture | | | Size After Hydroponic Culture | | | Size Growth Ratio | | |
|---|---|---|---|---|---|---|---|---|---|---|
| | | $L$/mm | $W$/mm | $T$/mm | $L'$/mm | $W'$/mm | $T'$/mm | $(L'-L)/L$ | $(W'-W)/W$ | $(T'-T)/T$ |
| 12 h | Jinong 1 | 9.21 | 8.62 | 7.41 | 9.59 | 9.37 | 8.60 | 4.13% | 8.70% | 16.06% |
| | Jinong 2 | 8.17 | 13.83 | 5.04 | 8.73 | 14.49 | 5.68 | 6.85% | 4.77% | 12.70% |
| | Jinong 3 | 12.19 | 11.32 | 9.28 | 12.49 | 11.91 | 9.74 | 2.46% | 5.21% | 4.96% |
| 24 h | Jinong 1 | 9.148 | 9.33 | 7.24 | 10.07 | 9.57 | 8.96 | 10.08% | 2.57% | 23.76% |
| | Jinong 2 | 7.99 | 13.35 | 6.17 | 8.34 | 14.20 | 7.26 | 4.38% | 6.37% | 17.67% |
| | Jinong 3 | 11 | 8.95 | 9.88 | 12.01 | 9.64 | 10.93 | 9.18% | 7.71% | 10.63% |
| 36 h | Jinong 1 | 10.03 | 7.45 | 7.89 | 10.61 | 9.59 | 9.70 | 5.78% | 28.72% | 22.94% |
| | Jinong 2 | 7.66 | 13.87 | 5.55 | 8.21 | 14.88 | 7.01 | 7.18% | 7.28% | 26.31% |
| | Jinong 3 | 11.51 | 11.29 | 9.13 | 12.46 | 11.80 | 10.63 | 8.25% | 4.52% | 16.43% |
| 48 h | Jinong 1 | 9.05 | 7.4 | 8.37 | 10.63 | 9.39 | 9.95 | 17.46% | 26.89% | 18.88% |
| | Jinong 2 | 7.55 | 15.81 | 4.90 | 8.85 | 17.40 | 7.93 | 17.22% | 10.06% | 61.84% |
| | Jinong 3 | 12.9 | 9.9 | 11.53 | 13.90 | 11.88 | 13.33 | 7.75% | 20.00% | 15.61% |

Then, Young's modulus of the seeds with different moisture content was measured by a universal testing machine; the detailed results are shown in Table 3. The results showed that Young's modulus of three varieties of seeds decreased with the increased moisture content.

**Table 3.** The Young's modulus of different moisture content for *Cyperus esculentus* seed.

| Jinong 1 | | Jinong 2 | | Jinong 3 | |
|---|---|---|---|---|---|
| Moisture Content | Young's Modulus /Mpa | Moisture Content | Young's Modulus /Mpa | Moisture Content | Young's Modulus /Mpa |
| 29% | 2.99 | 32% | 3.97 | 38% | 312 |
| 40% | 2.69 | 38% | 3.11 | 39% | 135 |
| 42% | 1.05 | 40% | 1.19 | 40% | 8.92 |
| 50% | 0.55 | 57% | 0.52 | 46% | 5.68 |

The point cloud data of the outlines of the *Cyperus esculentus* seeds were obtained using a Minolta Vivid 910 3D laser scanner (Minolta Co., Osaka, Japan) with an accuracy of 0.05 mm. Then, the scanned data were processed to the outer shape by using the reverse engineering software Geomagic Studio Wrap 2020, as shown in Figure 3. When filing spheres, the hilum of the seed was separately considered to be a single sphere. The other parts of the seed were represented as simplified ellipsoids. The outlines of the filing sphere were tangent to the outlines of the ellipsoid as far as possible.

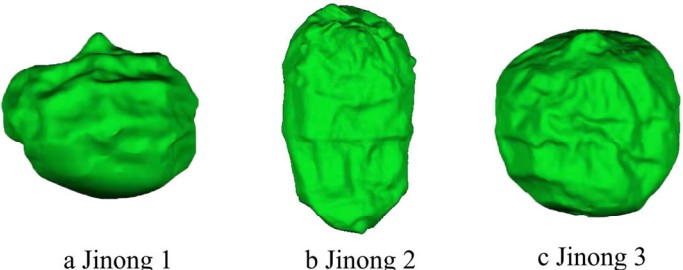

a Jinong 1      b Jinong 2      c Jinong 3

**Figure 3.** (**a–c**) The outer shapes of the *Cyperus esculentus* seeds.

Seven-, 9- and 11-sphere models of Jinong 1 were successively modeled, as shown in Figure 4a–c. 9-, 11- and 13-sphere models of Jinong 2 were successively modeled, as

shown in Figure 5a–c. 7-, 9- and 11-sphere models of Jinong 3 were modeled, as shown in Figure 6a–c. The detailed filing method is described in [22]. In addition, for comparison, 11-, 188-, and 1811-sphere models of Jinong 1 were developed based on the automatic filing function in EDEM software 2018, as shown in Figure 4d–f. Thirteen-sphere, 170- and 1650-sphere models of Jinong 2 were developed based on automatic filing, as shown in Figure 5d–f. Eleven-, 181- and 1782-sphere models of Jinong 3 were developed based on automatic filing, as shown in Figure 6d–f. The 188-sphere model of Jinong 1, the 170-sphere model of Jinong 2, and the 181-sphere model of Jinong 3 were automatically generated when the smoothness equaled 2. The 1811-sphere model of Jinong 1, the 1650-sphere model of Jinong 2, and the 1782-sphere model of Jinong 3 were automatically generated when the smoothness equaled 0. When the smoothness equaled 0, the MS model most closely simulated the real seed.

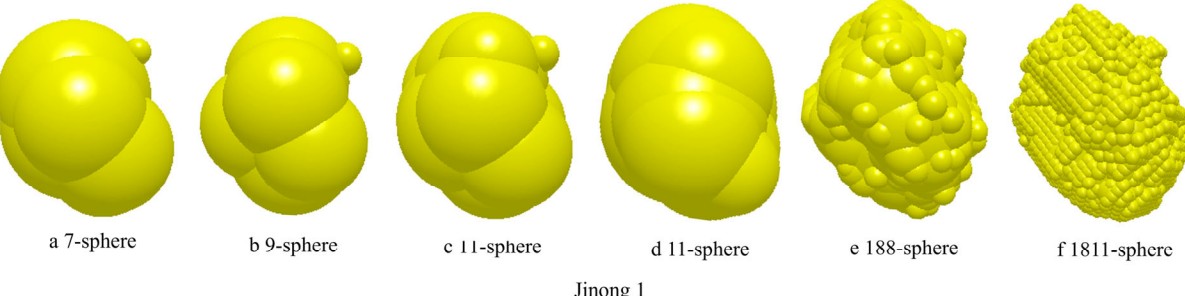

**Figure 4.** (**a**–**f**) MS models with different numbers of filing spheres for Jinong 1.

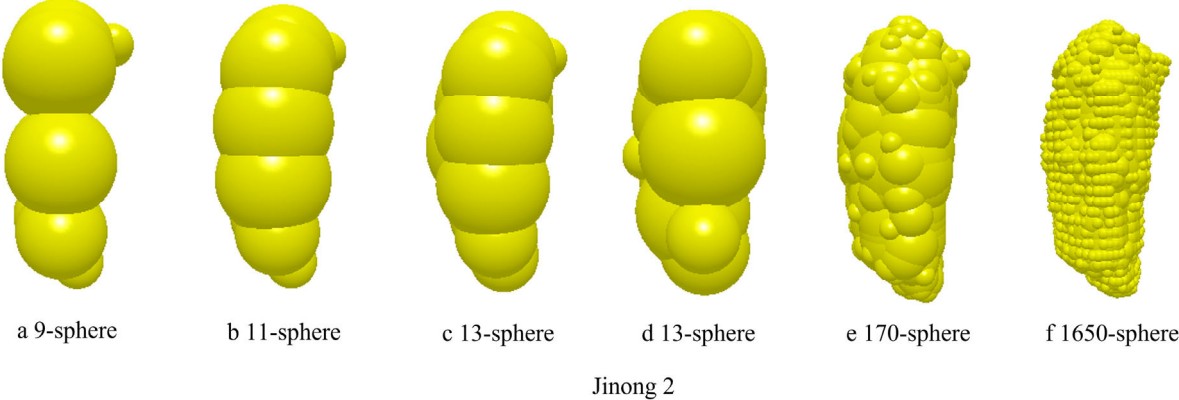

**Figure 5.** (**a**–**f**) MS models with different numbers of filing spheres for Jinong 2.

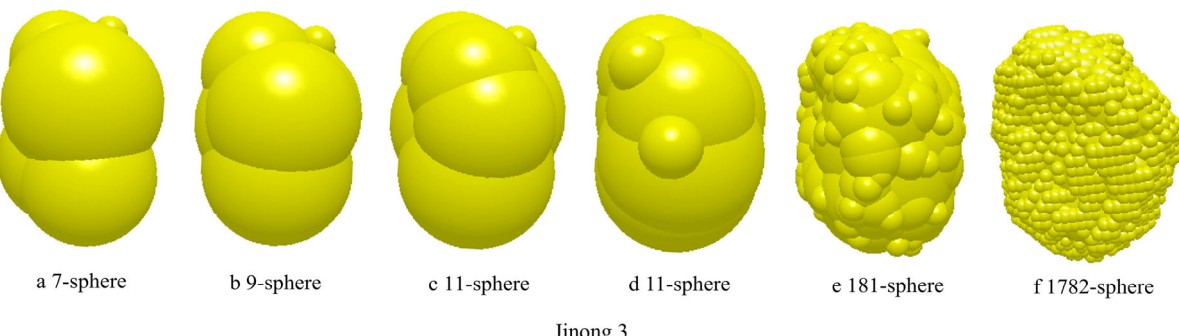

**Figure 6.** (**a**–**f**) MS models with different numbers of filing spheres for Jinong 3.

To quantify the accuracy of the MS model, we considered the volume ratio as an index which was defined as the volume of the MS particle model to the volume of the MS particle

model with the maximum number by automatic filing for each variety. Results shown in Table 4 indicate that the volume of the MS model, whether modeled by manual filing or automatic filing, was near the real seed particle shape. The number of filing spheres by manual filing was less than that by automatic filing.

**Table 4.** Volume and volume ratio for MS models with different numbers of filing spheres.

| Variety | Filing Method | Number of Filing Spheres | Volume/m$^3$ | Volume Ratio |
|---|---|---|---|---|
| Jinong 1 | Manual filing | 7-Sphere | $3.5746 \times 10^{-7}$ | 103.68% |
| | | 9-Sphere | $3.21057 \times 10^{-7}$ | 93.12% |
| | | 11-Sphere | $3.47738 \times 10^{-7}$ | 100.86% |
| | Automatic filing | 11-Sphere | $3.5734 \times 10^{-7}$ | 103.64% |
| | | 188-Sphere | $3.6420 \times 10^{-7}$ | 105.63% |
| | | 1811-Sphere | $3.4479 \times 10^{-7}$ | 100.00% |
| Jinong 2 | Manual filing | 9-Sphere | $3.0109 \times 10^{-7}$ | 85.79% |
| | | 11-Sphere | $3.5378 \times 10^{-7}$ | 100.80% |
| | | 13-Sphere | $3.7178 \times 10^{-7}$ | 105.93% |
| | Automatic filing | 13-Sphere | $3.7143 \times 10^{-7}$ | 105.83% |
| | | 170-Sphere | $3.8468 \times 10^{-7}$ | 100.00% |
| | | 1650-Sphere | $3.5097 \times 10^{-7}$ | 100.00% |
| Jinong 3 | Manual filing | 7-Sphere | $6.5727 \times 10^{-7}$ | 102.66% |
| | | 9-Sphere | $6.4211 \times 10^{-7}$ | 100.29% |
| | | 11-Sphere | $6.4029 \times 10^{-7}$ | 100.01% |
| | Automatic filing | 11-Sphere | $6.4789 \times 10^{-7}$ | 101.20% |
| | | 181-Sphere | $6.6884 \times 10^{-7}$ | 104.47% |
| | | 1782-Sphere | $6.4023 \times 10^{-7}$ | 100.00% |

## 3. Model Verification

To improve the reliability of the verification approach, the direct shear test, piling test, bulk density test, and rotating hub test were all considered. They were used to verify the accuracy of the modeled seed particles with different filing numbers. The test of each variety was repeated five times, and the mean value of the five experiments was used as the final value. Before each test, the sample used should be weighted. To guarantee the quality of the seeds assembly in simulation could keep the same as the ones in the test.

### 3.1. Physical Tests

3.1.1. The Direct Shear Test

A ZJ strain-controlled direct shear apparatus is shown in Figure 7. The experimental procedure was taken from [22]. Each specimen was loaded to 200 kPa normal compressive stresses. The experimental results were as follows: the shearing strength for Jinong 1 was 118.23 kPa; the shearing strength for Jinong 2 was 136.67 kPa; and the shearing strength for Jinong 3 was 105.76 kPa.

3.1.2. Piling Test

The experimental procedure was taken from [22]. The captured picture of the piling test and the binarized image are shown in Figure 8. The test results were as follows: the static angle of repose for Jinong 1 was 31.76° with a standard deviation of 1.15760°; the static angle of repose for Jinong 2 was 31.16° with a standard deviation of 1.8137°; and the static angle of repose for Jinong 3 was 31.83° with a standard deviation of 0.9446°.

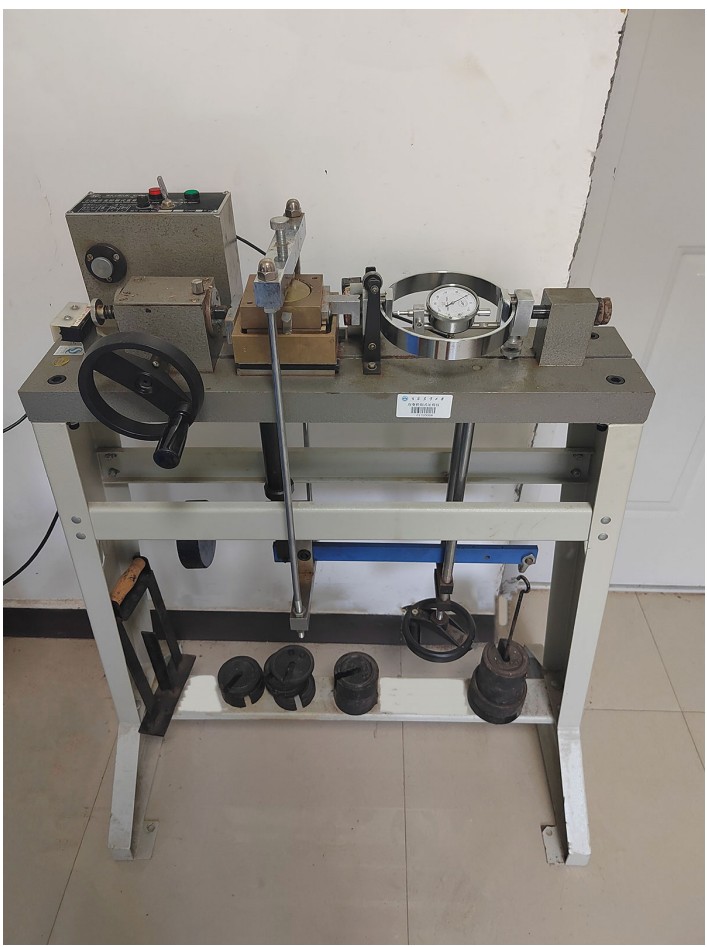

**Figure 7.** ZJ strain-controlled direct shear apparatus [22].

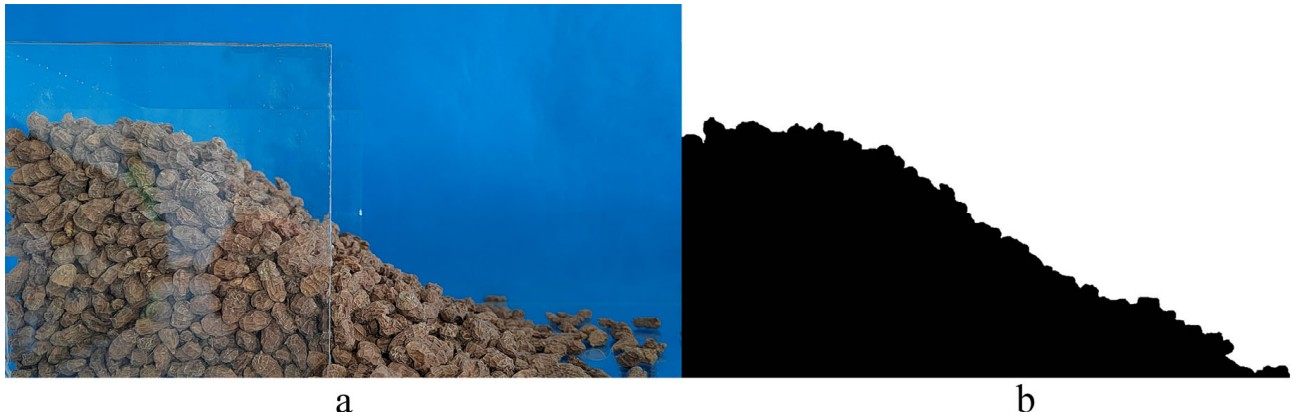

a                                                                                  b

**Figure 8.** Piling test: (**a**) photograph of the piling test; (**b**) image binarization [22].

3.1.3. Bulk Density Test

The *Cyperus esculentus* seeds were released above the box. When the seeds were stable, a scraper was used to scrape the redundant seeds, as shown in Figure 9. The test results were as follows: the bulk density of Jinong 1 was 680.75 g/m$^3$ with a standard deviation of 17.27 g/m$^3$; the bulk density of Jinong 2 was 664.62 g/m$^3$ with a standard deviation of 14.41 g/cm$^3$. The bulk density of Jinong 3 was 609 g/cm$^3$ with a standard deviation of 9.11 g/cm$^3$.

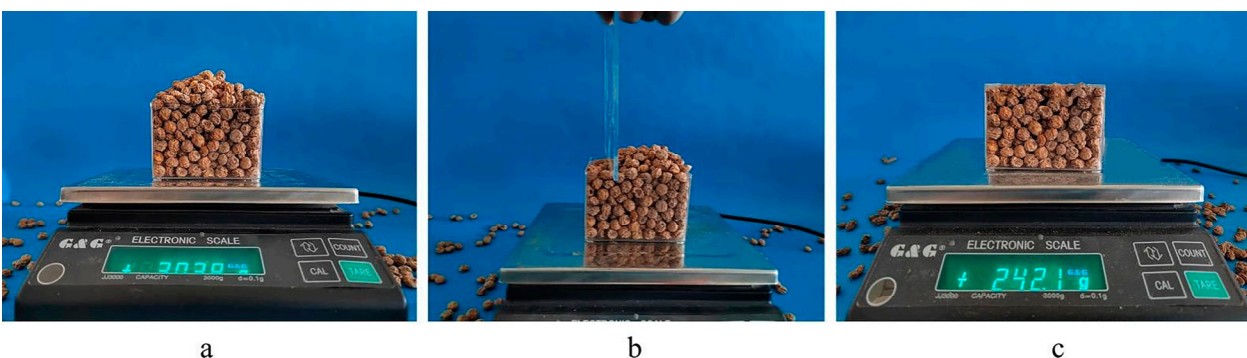

**Figure 9.** Bulk density test of the *Cyperus esculentus* seeds [22]. (**a**) particle assembly accumulation; (**b**) use the scratch board to make the surface flat; (**c**) measuring the residual mass of the seeds.

### 3.1.4. Rotating Hub Test

A rotating hub apparatus was used for the rotating hub test, as shown in Figure 10a. The inside diameter and thickness of the steel rotating hub were 200 mm and 50 mm, respectively. A high-speed camera was employed for recording. The accumulation behavior of irregular particles was complex. Therefore, according to Formulas (1) and (2), the particle filing rate was 11.04%.

$$f = \frac{1}{\pi}(\varepsilon - sin\varepsilon cos\varepsilon),\tag{1}$$

$$\varepsilon = arccos\left(1 - \frac{h}{RC}\right),\tag{2}$$

where $f$ is the filing rate; $\varepsilon$ is the angle of filing; $h$ is the height of the particle assembly; and $R_c$ is the radius of the rotating hub.

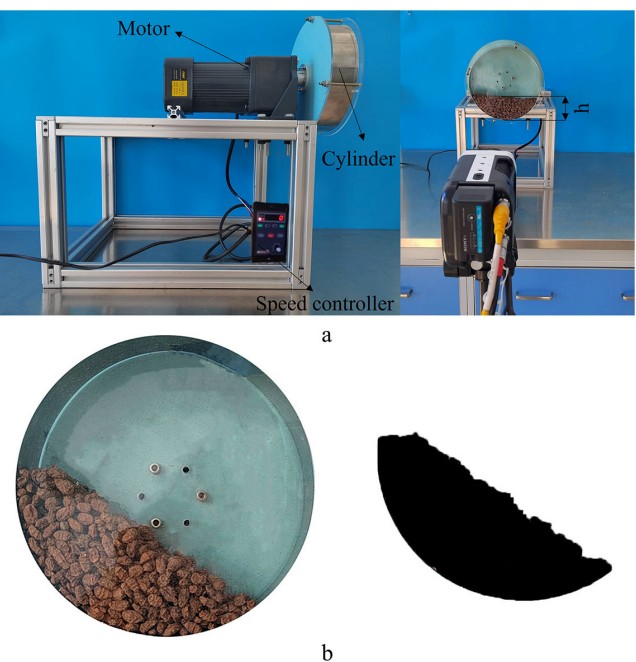

**Figure 10.** Rotating hub test: (**a**) photograph of the rotating hub test; (**b**) image binarization.

According to Formula (3) [30], the rotating speed $\omega_c$ of the rotating hub was selected as 5, 10, and 15 rpm:

$$w_c = \sqrt{\frac{gFr}{Rc}},\tag{3}$$

where $\omega_c$ is the rotating speed; $F_r$ is the Froude number; and $F_r = 1 \times 10^{-4} - 1 \times 10^{-2}$ [31].

Flow images of seed particles at 3 s, 4 s, and 5 s were selected, and binary processing was achieved, as shown in Figure 10b. The dynamic angle of repose was measured, and the mean was considered to be the final value. The test results are shown in Figure 11. According to the results, with increasing rotation speed, the dynamic angle of repose of the seeds increased for different varieties.

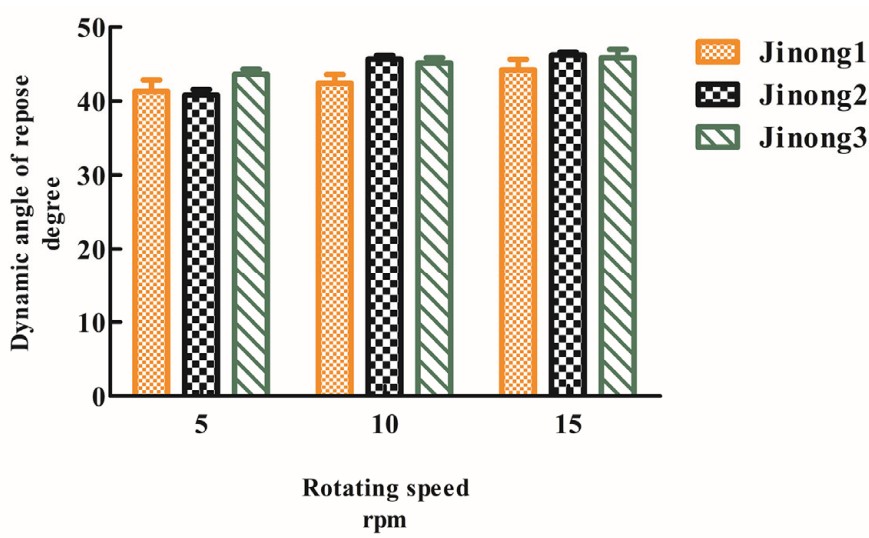

**Figure 11.** Dynamic angle of repose for the *Cyperus esculentus* seeds.

*3.2. Simulation Analysis*

The direct shear test, piling test, bulk density test, and rotating hub test were both simulated. The simulations were performed using 7-, 9-, and 11-sphere models in manual filing and using 11- and 188-sphere models in an automatic filing with Jinong 1. 9-, 11-, and 13-sphere models in manual filing and 13- and 170-sphere models in the automatic filing were performed with Jinong 2. In addition, 7-, 9-, and 11-sphere models in manual filing and 11- and 181-sphere models in the automatic filing were performed with Jinong 3.

The version of EDEM 2018, DEM Solutions, Edinburgh, UK, 2002 was used in the simulation. The HM contact model was selected in the simulation. The time step in the simulations was $5 \times 10^{-7}$ s. The *Cyperus esculentus* seeds were generated by a normal distribution in terms of volume. The mean and the standard deviation of the width for each type are listed in Table 5. Relevant simulation parameters are listed in Table 6. The Poisson's ratio of *Cyperus esculentus* seeds was obtained from the standard of the American Society of Agricultural Engineers. The density of *Cyperus esculentus* seeds was measured in Section 1. The coefficient of restitution of the seeds was obtained by a single pendulum impact test, and the coefficient of restitution between seeds and contact materials was obtained by a free drop test. The coefficient of friction between seeds and contact materials was determined by the slope method. The coefficients of static friction and rolling friction between the *Cyperus esculentus* seeds were obtained from the Plackett–Burman test combined with the path of the steepest ascent method through the direct shear test. The detailed procedures are described in [22].

**Table 5.** The mean and the standard deviation of the width of different varieties of *Cyperus esculentus* seeds.

| Variety | Mean/mm | Standard Deviation/mm |
|---|---|---|
| Jinong 1 | 9.1098 | 1.1916 |
| Jinong 2 | 13.695 | 1.7046 |
| Jinong 3 | 11.452 | 1.5552 |

**Table 6.** Simulation parameters of the tests [22].

| Parameter | Jinong 1 | Jinong 2 | Jinong 3 | Polymethyl Methacrylate | Steel | Copper |
|---|---|---|---|---|---|---|
| Poisson's ratio | 0.4 | 0.4 | 0.4 | 0.32 | 0.3 | 0.35 |
| Density, kg/m$^3$ | 1340 | 1270 | 1190 | 1190 | 7850 | 8960 |
| Young's modulus, MPa | 165 | 165 | 165 | 1197 | 20,000 | 100 |
| Coefficient of restitution | 0.45 | 0.45 | 0.45 | 0.55 | 0.65 | 0.5 |
| Coefficient of static friction | 0.55 | 0.35 | 0.35 | 0.34 | 0.4 | 0.4 |
| Coefficient of rolling friction | 0.05 | 0.05 | 0.05 | 0.05 | 0.05 | 0.05 |

### 3.2.1. Direct Shear Test Simulation

The direct shear test simulation procedure was kept the same as that of the real tests. Different models of seeds were considered in simulations. The screenshots of the simulation for different times are shown in Figure 12. The shearing strength was determined as the test index, and each simulation was repeated three times. The mean was the final value, and the results are shown in Figure 13.

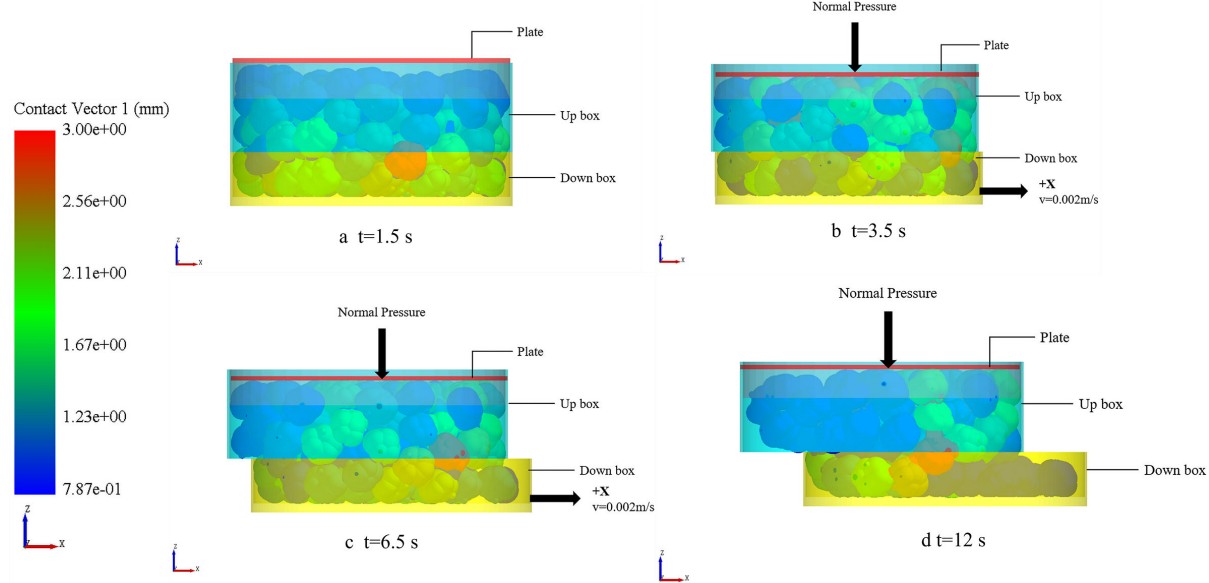

**Figure 12.** Snapshots of the simulation of the direct shear test of Jinong 1 seeds using the 9-sphere model [22].

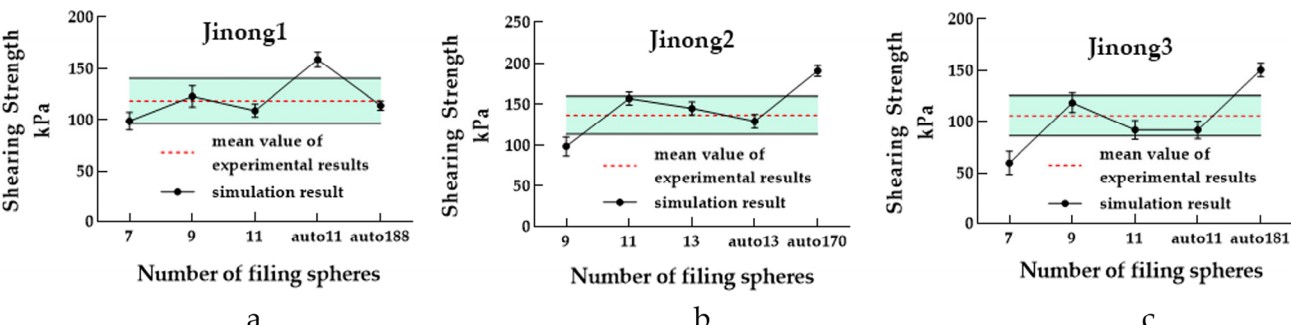

**Figure 13.** Simulation results of the direct shear test.

The experimental and simulated results of the shearing strength of particle models with different filing spheres for three varieties of seeds are shown in Figure 13. Except for the 9-sphere of Jinong 2 and the 7-sphere of Jinong 3, the simulated results of the MS models by manual filing converged to the mean and fluctuated within the standard deviation of

the experimental results. The simulated results of the 11-sphere model with Jinong 1, the 170-sphere model with Jinong 2, and the 181-sphere model with Jinong 3 created by automatic filing were not closer to the mean of the experimental results. These results thus showed that increasing the number of filing spheres could not effectively improve the simulation accuracy. In contrast, the CPU computation time increased with an increasing number of filing spheres, as shown in Figure 14.

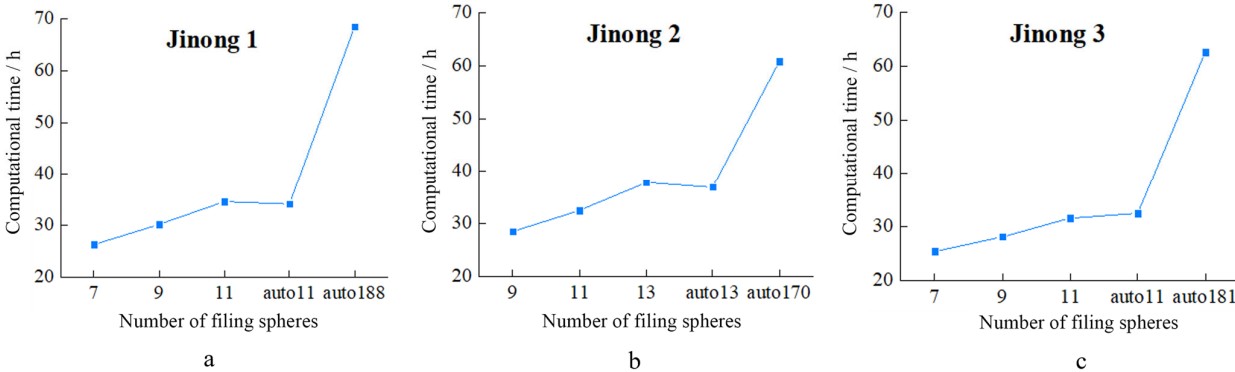

**Figure 14.** Computational times of the direct shear test simulation.

### 3.2.2. Piling Test Simulation

The piling test simulation procedure was kept the same as that of the real tests. Different models of seeds were considered in simulations. The screenshots of the simulation for different times are shown in Figure 15. The static angle of repose was determined as the test index, and each simulation was repeated three times. The mean was the final value, and the results are shown in Figure 16.

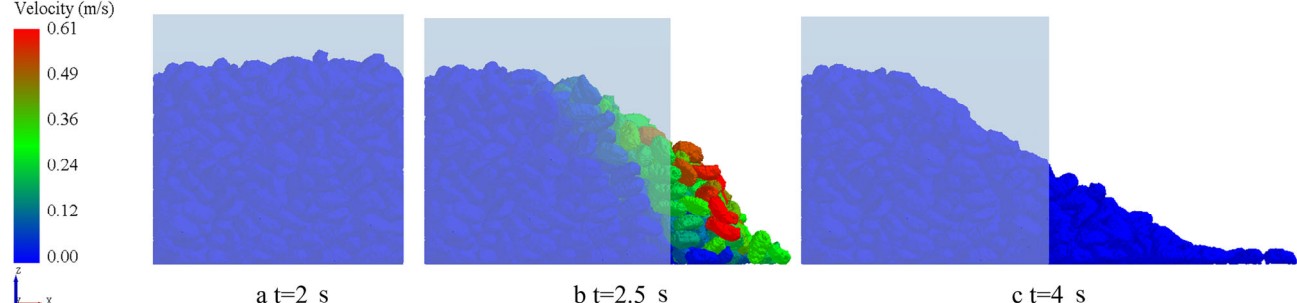

**Figure 15.** Snapshots of the simulation of the piling test of Jinong 2 seeds using the 170-sphere model.

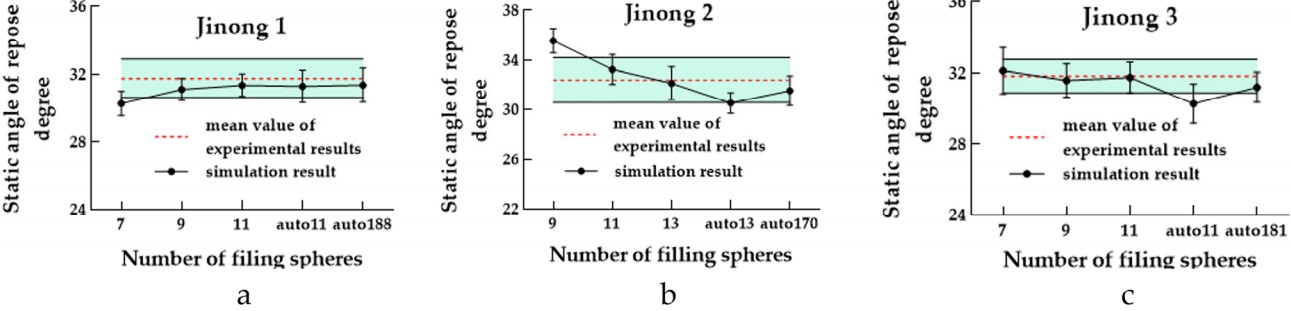

**Figure 16.** Simulation results of the piling test.

The experimental results and the simulated results of the static angle of repose of particle models with different filing spheres for three varieties of seeds are shown in Figure 16. Except for the 7-sphere model with Jinong 1 and the 9-sphere model with

Jinong 2, the simulated results of the MS models by manual filing converged to the mean and fluctuated within the standard deviation of the experimental results. Except for the 11-sphere of Jinong 3 created by automatic filing, which was not closer to the mean of the experimental results, the other automatic filing models converged to the mean of the experimental results. The CPU computation time increased with the increasing number of filing spheres, as shown in Figure 17.

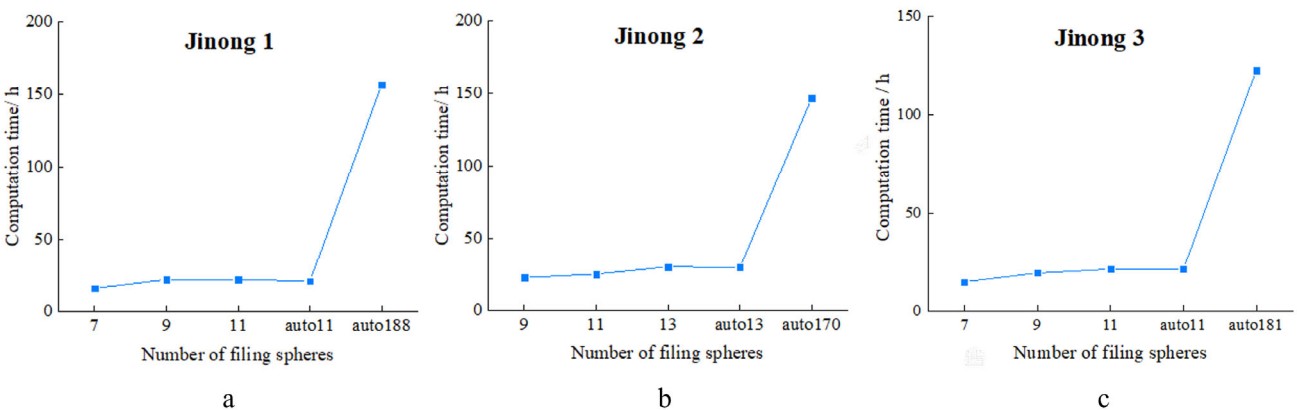

a

b

c

**Figure 17.** Computational times of the piling test simulation.

### 3.2.3. Bulk Density Test Simulation

The bulk density test simulation procedure was kept the same as that of the real tests. Different models of seeds were considered in simulations. The screenshots of the simulation for different times are shown in Figure 18. The bulk density was determined as the test index, and each simulation was repeated three times. The mean was the final value, and the results are shown in Figure 19.

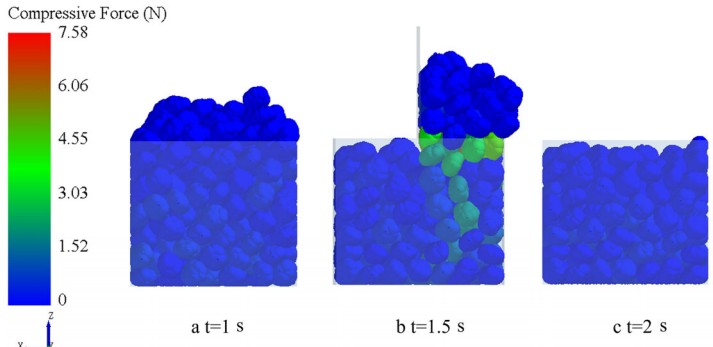

**Figure 18.** Snapshots of the simulation of the bulk density test of Jinong 3 seeds using the 11-sphere model.

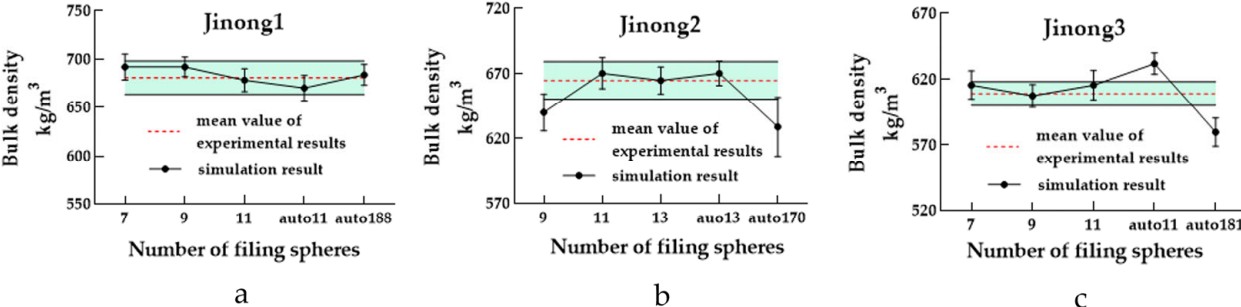

a

b

c

**Figure 19.** Simulation results of the bulk density test.

The experimental and simulated results of the bulk density of particle models with different filing spheres for three varieties of seeds are shown in Figure 19. The mean of the simulated results of MS models by manual filing converged to the mean and fluctuated within the standard deviation of the experimental results. Except for the 170-sphere model with Jinong 2, the 11- and 181-sphere with Jinong 3 created by automatic filing were not closer to the mean of the experimental results, and the other automatic filing models converged to the mean of the experimental results. Additionally, the CPU computation time was considered and increased with an increasing number of filing spheres, as shown in Figure 20.

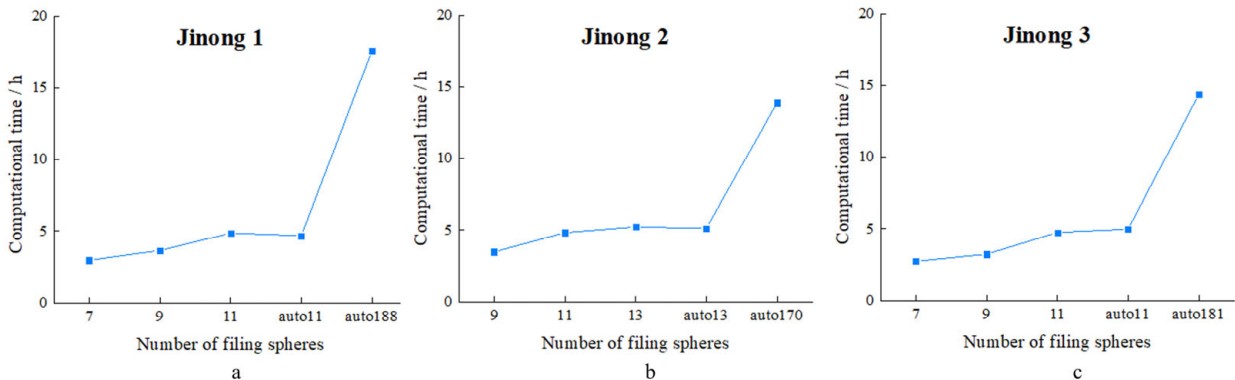

**Figure 20.** Computational times of the bulk density test simulation.

### 3.2.4. Rotating Hub Test Simulation

The rotating hub test simulation procedure was kept the same as that of the real tests. Different models of seeds were considered in simulations. The snapshots at 2 s, 3 s, and 4 s of the simulation were captured, as shown in Figure 21. The dynamic angle of repose was determined as the test index, and each simulation was repeated three times. The mean was the final value, and the results are shown in Figure 22.

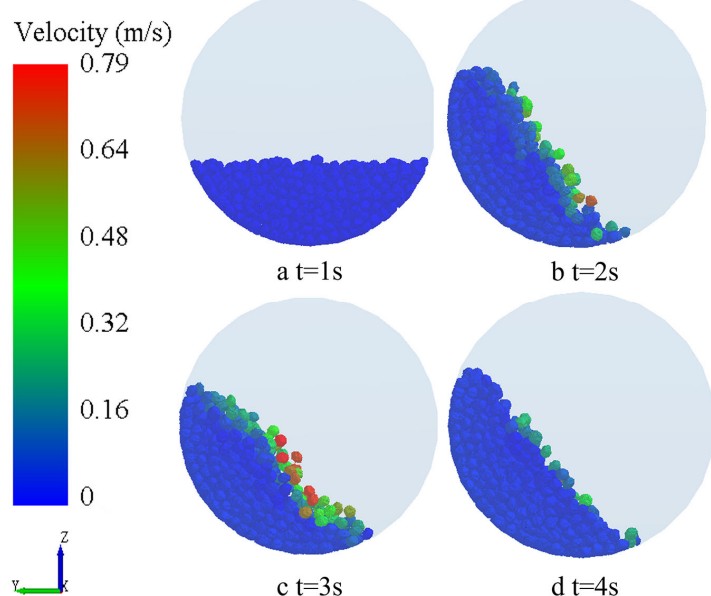

**Figure 21.** Snapshots of the simulation of the rotating hub test of Jinong 1 seeds using the 188-sphere model.

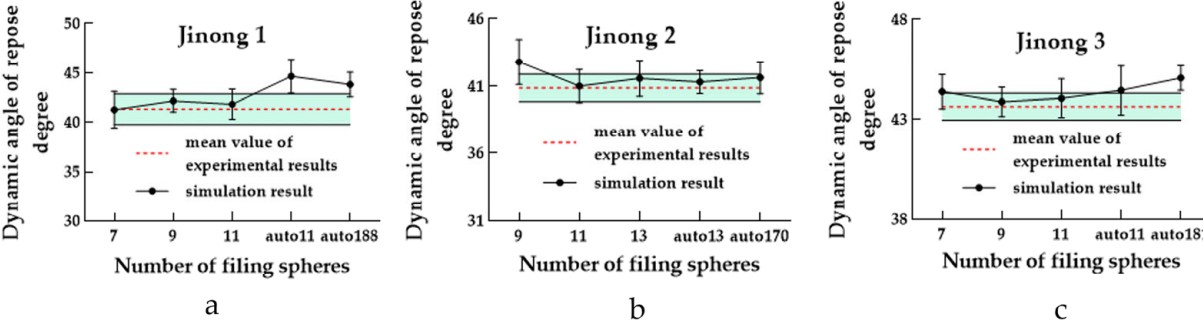

**Figure 22.** Simulation results of the rotating hub test.

The experimental and simulated results of the dynamic angle of repose of particle models with different filing spheres are shown in Figure 22. Except for the 9-sphere model with Jinong 2, the mean of the simulated results of the MS models by manual filing converged to the mean value and fluctuated within the standard deviation of the experimental results. Except for the 11- and 188-sphere models with Jinong 1 and the 181-sphere model with Jinong 3 created by automatic filing, which was not closer to the mean of the experimental results, the other automatic filing models converged to the mean of the experimental results. Additionally, the CPU computation time was considered and increased with an increasing number of filing spheres, as shown in Figure 23.

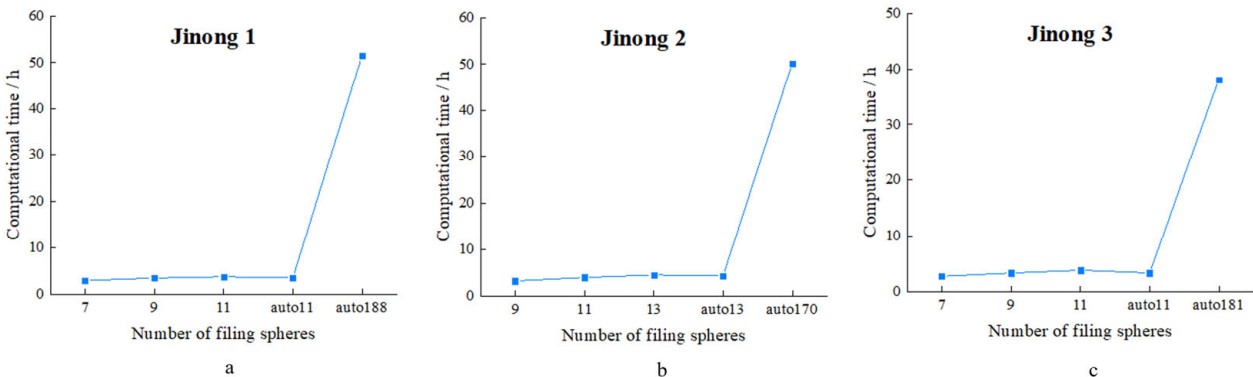

**Figure 23.** Computational times of the rotating hub test simulation.

According to the simulation results, the 9-sphere model with Jinong 1, the 11-sphere model with Jinong 2, and the 9-sphere model with Jinong 3 could provide a good compromise between simulation accuracy and time consumed.

## 4. Multicontact Points Issue

The HMNR model could effectively reduce the influence of the multicontact points issue when using MS modeling in the simulation [24,29]. Therefore, the HM model and HMNR model were both used for comparison. Taking Jinong 1 as an example, the effects of multicontact points on the movement of particle assembly and the movement of individual particles were studied through the rotating hub test and free drop test. Additionally, the influence of Young's modulus on the number of contact points was studied with a free drop test.

### 4.1. Influence of Multicontact Points on Particle Assembly Movement

Taking the 188-sphere model with Jinong 1 as an example, simulation tests were conducted using the HM model and HMNR model through the rotating hub test. The rotating speed was 5 rpm. All the simulation parameters were consistent, as shown in Table 6. The simulation screenshots are shown in Figure 24. Each simulation was repeated

three times, and the mean was reported as the final result. The dynamic angle of repose was 43.80° with a standard deviation of 7.46° using the HM model. The dynamic angle of repose was 43.75° with a standard deviation of 5.68° using the HMNR model. The difference in the dynamic angle of repose between the two results by different contact models was only 0.11%. The influence of multiple contact points on particle assembly movement was insignificant.

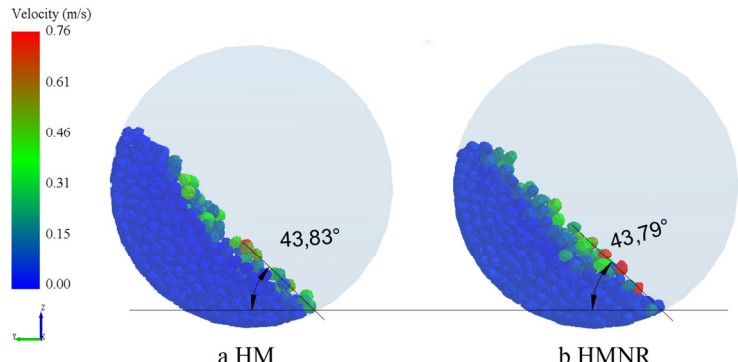

**Figure 24.** Snapshots of the simulation for the rotating hub test with the different contact models.

*4.2. Influence of Multicontact Points on Individual Particle Movement*

Taking the 188-sphere model with Jinong 1 as an example, simulation tests were conducted using the HM model and HMNR model through the free drop test. The release height was 150 mm. All the simulation parameters were consistent, as shown in Table 6. Because particles were randomly generated from the particle factory, each test was repeated 10 times. The simulation result is shown in Figure 25. Particle 1 was generated using the HMNR model, and Particle 2 was generated using the HM model. The mean of the rebounding height of Particle 1 was 58.00 mm with a standard deviation of 6.76 mm. However, the mean of the rebounding height of Particle 2 was 41.50 mm with a standard deviation of 8.36 mm. The maximum difference in the rebounding height was 40.26 mm. The influence of multiple contact points on individual particle movement was significant.

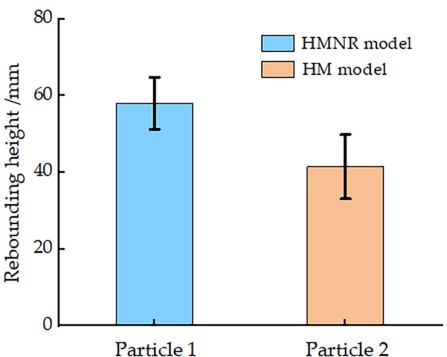

**Figure 25.** Simulation result for the free drop test with the different contact models.

*4.3. Influence of Young's Modulus on Multiple Contact Points*

Taking the 188-sphere model with Jinong 1 as an example, simulation tests were conducted using the HMNR model through the free drop test. The release height was 150 mm. Taking Young's modulus as the variable, Young's modulus was set as in [22]. The Young's moduli were determined to be $1.65 \times 10^6$ Pa, $1.65 \times 10^7$ Pa, and $1.65 \times 10^8$ Pa, respectively.

The rebounding height and contact points were counted with different values of Young's modulus. Each simulation was repeated 10 times, and the mean value was the final result. The results are shown in Figure 26. The contact points decreased with increasing Young's modulus. In contrast, the rebounding height increased with increasing Young's

modulus. These results showed that the change in Young's modulus had a certain effect on the number of contact points.

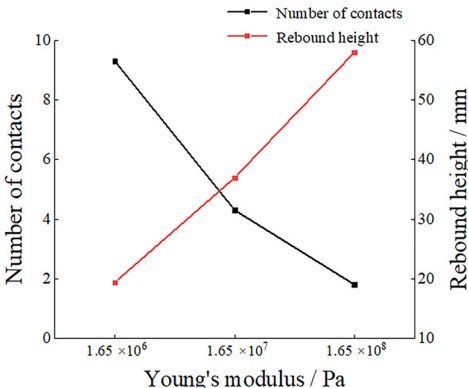

**Figure 26.** Relationship between the bounce height, the number of contacts and Young's modulus.

## 5. Conclusions

In this study, DEM models of *Cyperus esculentus* seeds were created with different numbers of filing spheres. Additionally, other particle models by automatic filing in EDEM software were also introduced for comparison. Four varieties of tests were used to verify the feasibility of the particle modeling of *Cyperus esculentus* seeds that we proposed. In addition, the influence of the multicontact points on the movement behavior of the particle assembly model and individual particle model was studied through simulation analysis. Finally, the influence of Young's modulus on the multicontact points was also investigated. The following conclusions are based on the results of this study:

(1) The direct shear test, piling test, bulk density test, and rotating hub test were used to verify the created DEM models with different numbers of filing spheres. Through comparisons between the experimental results and the simulated results, combined with CPU computational time, the DEM modeling created in this paper could achieve better simulation accuracies with fewer filing spheres. According to the simulation results, the 9-sphere model with Jinong 1, the 11-sphere model with Jinong 2, and the 9-sphere model with Jinong 3 could provide a good compromise between simulation accuracy and time consumed.

(2) According to the simulation results, some limitations were present when using one single verification test, and the differences in DEM models with different numbers of filing spheres could not be fully described. Therefore, to improve the reliability of the verification approach and guarantee the accuracy of the model, more varieties of tests are required. In addition, the precision particle shape model could not markedly improve the simulation accuracy. In contrast, the greater the number of filing spheres included, the longer the CPU calculation time consumed.

(3) Comparing the HMNR model with the HM model, the influence of the multicontact points on the movement behavior of the particle assembly model and individual particle model based on the MS method was studied through simulation analysis of the rotating hub test and the free drop test. According to the simulation results, the multicontact points affected the motion behavior of individual particles. In contrast, the influence of multiple contact points on the movement behavior of particle assembly was insignificant.

(4) In addition, the relationships between the moisture content of seeds and Young's modulus, Young's modulus, and the number of contact points were also studied in simulations and experiments; with an increasing Young's modulus, the number of contact points decreased, while the rebound height increased when creating the particle model based on the MS method. According to the experimental results,

with increasing moisture content, Young's modulus decreased, while the tri-axial dimensions increased.

**Author Contributions:** Conceptualization, J.W. and T.X.; methodology, R.Z.; software, T.X. and Y.W.; validation, X.J. and R.Z.; formal analysis, T.X.; investigation, R.Z.; resources, W.F. and R.Z.; data curation, Y.W., W.F. and R.Z.; writing—original draft preparation, T.X.; writing—review and editing, T.X.; visualization, X.J.; supervision, J.W.; project administration, J.W.; funding acquisition, X.J. All authors have read and agreed to the published version of the manuscript.

**Funding:** This research was funded by Jilin Provincial Science and Technology Development Plan Project, No. 20200402008NC.

**Data Availability Statement:** The study did not report any data.

**Acknowledgments:** In this paper, we received technical support from Jilin University, including the Licensed software of EDEM.

**Conflicts of Interest:** The authors declare no conflict of interest.

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
