# Peer review of "Study on Verification Approach and Multicontact Points Issue When Modeling Cyperus esculentus Seeds Based on DEM"

_processes, doi:10.3390/pr11030825_

Round 1
Reviewer 1 Report
The discrete element method (DEM) has become widely used in many application fields. In this manuscript the authors studied three varieties of Cyperus esculentus seeds or tubers by using DEM Simulation with multisphere models. As these models are generated with spheres, the more spheres used, the higher the apparent precision of the model, but also the computational cost of the simulations. Finding the balance between these two components is always a difficult task. I believe that this type of work, even without being very original, helps us to contribute to the knowledge of granular materials.
I believe some issues should be addressed but then would have no problems in recommending this article for publication.
Title: fundamental question on this paper: are you studying seeds or tubers?l.13: You do not establish particles, you model or create particles, right? You have used this word 18 times.
There are no sense sentences: Just to mention a few:Second paragraph (l.45 to l.84)
l.60: Therefore, how to establish a more precise and appropriate DEM model for Cyperus esculentus seeds should be studied in more detail. In addition, the established particle model must be further verified.
l.64: However, how to verify the established DEM model and improve the reliability of the verification methods must be studied in more detail.
l.67: The multicontact points could lead to a contact that is too stiff in the simulation, accompanied by a system with excessive damping
l.206: Because the rotating speed and filing rate influenced the flow conditions of seeds in the rotating hub.
l.384: The results are shown in Fig. 26. As shown in Fig. 26
l.100: How many seeds of the three varieties of Cyperus esculentus did you use to measure?
l.109-112: There is something wrong with these two sentences.
Table 2: the column "size before" is a control sample?
the column "size growth ratio" should not have dimensions (L instead of L/mm)
l.140: How do you perform the point cloud data?
l.141: how do you perform the "filing" of the particles?
Table 4: Wrong units: "Volume kg/m³"
l.175: you do not improve the reliability of the test but the model, right?
Physical tests: General:
How many repetitions of each test did you perform?
How many samples did you analize?
Which are the dimensions of each test apparatus?
Shear:
errors?
Since one has only one layer of grains: Is the shear apparatus used to measure this type of sample appropriate?
Pilling:
What was the measurement procedure? It has been seen that these tests depend on the way in which the measurement is carried out.
The precision of the standard deviation is not appropriate.
Rotating hub:
There is no clear introduction of the formulas used.
What is the Froude number?
Why the Froude number is Fr =1×102-1×104.
l.232: "Cyperus esculentus seeds were generated by a normal distribution in terms of volume." Which is the mean and the standard deviation of the distribution?
How many particles did you use in each simulation?
l.248: "each simulation was repeated three times" Why? Error values?
Fig. 12, there are very few layers of particles! (almost a monolayer!)
In Fig. 14, for the same number of particles, the computational cost of J1 & J2 in the auto mode is less than the manual. Why don't you try to use auto mode with less particle cases (7,9 for J1 and 9, 11 for J2)
Figures 14 and 17 are the same?
Fig.21, Which is the slope here?? There are several possible measures depending on the time you look at it.
The discussion section seems to be part of the results.
Fig.25, This figure is not clear. Maybe you can graph the average value of the height of the first n bounces for each case.
Fig.26, error bars?
Conclusion (1): "could achieve better simulation accuracies with fewer filing spheres"
Why then do the authors not use between one and 6 particles to compare?
Conclusion (2): what do they mean by: "some limitations were present when using one single verification test"
and with: "the differences in DEM models with different numbers of filing spheres could not be fully described."
"to improve the reliability of verification" or the reliability of the particle model?
Author Response
Dear Reviewer, Thank you very much for your advice. We have revised the manuscript, and would like to re-submit it for your consideration. We have addressed the comments and the amendments are highlighted in red in the revised manuscript. Point by point responses to your comments are listed below this letter. We would like to express our sincere thanks to you for the constructive and positive comments. We hope that the revised version of the manuscript is now acceptable for publication. I look forward to hearing from you soon. With best wishes, Yours sincerely, Tianyue Xu First author
Reviewer 2 Report
The paper presents interesting topics providing rigorous and robust analysis but presentation can be improved a bit, especially that discussion section as conclusion is not the place to provide your discussion.
Some experimental details are missing, e.g. how seeds dimensions are measured, line 136: provide details on the universal testing machine.
Your discussion section is introducing new results rather than discussion previously presented results, e.g. trying to provide some general trends in data across different test and some general guidelines for the future research (e.g. it seems 9 or 11 might be a good compromise between simulation accuracy and time demand across all results). Later you give this in your conclusion but this is not discussed in detail before, results cannot introduce any new ‘conclusion’ just summarize the main point.
Another point on your discussion section, ie. multicontact points study, why 88-sphere model was used as not the best choice in the previous section? Why not manual filled seeds are used?
Fig. 3 looks really nice but missing details how it was measured and produced.
Please provide a bit more details on your piling test, ie. how it was performed.
Section 3.1.4, was the filling ration 11.04% in all the experiments?
Your rotating speed was not ‘chosen’ by eq. 3, but the question is why you have chosen to have that specific range of Fr number? Any specific reason?
Lots of typesetting issues, e.g. chapter 2 title is an orphan, subsections 3.2.3 & 4.3 as well; figure 2 caption not at the same page, also not sure why text is wrapped around it, table 4 over two pages,
Author Response

(The authors gave the same response as above.)

Round 2
Reviewer 1 Report
The authors have corrected some points, but still have some issues to solve:
There is a new author, why?
It seems to me that it should be properly clarified.
Minor issues:
l. 28- “Moreover, The relationships” (the capital letter in “The”).
l. 30- “moisture content. And the number”. The sentence begins with “And”?
l. 55- “modeled a corn seed model based on” change to “modeled a corn seed based on”
l. 180- “modeled seed particle models with” change to “modeled seed particles with”
l. 196- “The specific experimental procedure is consistent with the literature [22].” It seems to me that the word you should use is not “consistent”, instead it could be: The experimental procedure was taken from [22]
l. 226- “seeds increased for different varieties of seeds.” or “seeds increased for different varieties.”
l. 365- “Also, the influence of Young's modulus on the number of contact points was also studied” there is a redundant “also”
l. 448- “(4) In addition, The relationships” The capital letter should not be there.
l. 450- “and experiments. with an increasing” the punctuation is not correct.
Important issues:
l. 188-191- The precision of the standard deviations are not appropriate.
l. 110- “Because the tri-axial dimensions and Young's modulus of seed particles were greatly
affected by moisture content.” This sentence is not correct.
l. 186- Shear test, it seems to me that you should at least mention that the standard conditions for the experiment are not being met.
l. 216-226. There is still no clear introduction to what you want to calculate and why. This entire paragraph is unclear. I know what the Froude number is and I see that you added an article, but there should be at least an intuitive explanation of it.
Point 11: It seems to me that they should add table 2 of the explanations in the manuscript clarifying that these are the mean and sd values that they used.
Point 12: The question is why 3 and not 5, 8 or 10?
Point 13: As in line 186, it seems to me that you should at least mention that the standard conditions for the experiment are not being met.
Point 14: What does it mean that “the modeling accuracy must be satisfied”. The question was: if the automatic filling mode of EDEM seems to be more effective computationally, why didn't you try the automatic filling with less particles (cases 7, 9 particles).
Point 16: You have to explain the method with which you measure the first time you use it.
Point 17: No changes here: sections 4.1 and 4.2 are results, not discussion.
Point 20. As in point 14, the accuracy, as defined between lines 170 and 172, is a ratio between volumes. Under this definition, one could generate the desired accuracy with an arbitrary number of particles, right?
Author Response
Dear Reviewer,
Thank you very much for your advice. We have revised the manuscript carefully, and would like to re-submit it for your consideration. We have addressed the comments and the amendments are highlighted in red in the revised manuscript. Point by point responses to your comments are listed below this letter. We would like to express our sincere thanks to you for the constructive and positive comments.
We hope that the revised version of the manuscript is now acceptable for publication.
I look forward to hearing from you soon.
With best wishes,
Yours sincerely,
Tianyue Xu
First author
